# Bacterial sexually transmitted infections and related antibiotic use among individuals eligible for doxycycline post-exposure prophylaxis in the United States

Anna M. Parker [1], Jennifer J. Chang[2], Ligong Chen[3], Laura M. King[1], Sandra I. McCoy[1,4], Joseph A. Lewnard [1,5,6,8] ✉ & Katia J. Bruxvoort [7,8]

While doxycycline postexposure prophylaxis (doxyPEP) can prevent bacterial sexually transmitted infections (STIs), concern surrounds the volume of antibiotic use needed to realize this benefit. We estimated incidence rates of gonorrhea, chlamydia, and syphilis diagnoses and related antibiotic prescribing among US males and transgender individuals using Merative MarketScan® Research Databases during 2017-2019. Follow-up encompassed 38,543 person-years among recipients of HIV pre-exposure prophylaxis (PrEP), 29,228 person-years among people living with HIV (PLWH), and 19,918 person-years among people with prior-year STI diagnoses. Incidence rates of STI diagnoses among PLWH and PrEP recipients with ≥1 prior-year STI diagnosis totaled 33.3-35.5 per 100 person-years. Direct effects of doxyPEP could prevent 7.4-9.6 gonorrhea diagnoses, 7.3-8.1 chlamydia diagnoses, and 3.1-5.9 syphilis diagnoses per 100 person-years of use. However, expected increases in tetracycline consumption resulting from doxyPEP implementation totaled 271.9-312.9 additional 7-day doxycycline treatment courses per 100 person-years of use. These increases corresponded to 37.0-38.7, 36.5-37.0, and 46.1-100.2 additional 7-day doxycycline treatment courses for each prevented gonorrhea, chlamydia, and syphilis diagnosis, respectively. Increases in doxycycline use exceeded anticipated reductions in STI-related prescribing of cephalosporins, macrolides, and penicillins by 16–69-fold margins. Anticipated changes in antibiotic use as well as STI incidence should inform priority-setting for doxyPEP.

Bacterial sexually transmitted infections (STIs) pose a substantial burden due to their prevalence and potential sequelae. In the United States (US), >2.4 million gonorrhea, chlamydia, and syphilis cases were reported in 2023, and the number of reported syphilis cases was 80% higher than in 2018[1]. Young people, men who have sex with men (MSM), transgender women (TGW), and people living with HIV (PLWH) experience disproportionate STI burden[2,3].

The US Centers for Disease Control and Prevention (CDC) recommended doxycycline post-exposure prophylaxis (doxyPEP) for prevention of bacterial STIs among certain populations in 2024 on the basis of shared clinical decision-making between patients and their

healthcare providers[4]. These recommendations encompass MSM and TGW who were diagnosed with a bacterial STI in the preceding 12 months. Doxycycline is typically well-tolerated and used for many acute bacterial infections[5]. Longer-term courses are also prescribed for malaria prophylaxis and for skin conditions, including moderate-to-severe inflammatory acne. DoxyPEP consists of 200 mg doxycycline taken orally within 72 h (and ideally <24 h) after sex[6]. Recent trials have demonstrated that doxyPEP reduces the incidence of chlamydia and syphilis infections among MSM, TGW, and PLWH by approximately 70%[6–8]. Results were less promising for gonorrhea infections, with efficacy estimates spanning 17% in a French trial[6] to 55% in a US trial[7].

However, concerns surround possible contributions of doxyPEP to increased antibiotic use and antimicrobial resistance (AMR)[9]. In the US, approximately half of all *Neisseria gonorrhoeae* infections are resistant to ≥1 antibiotic[10]; models suggest broad doxyPEP implementation may exacerbate this threat[11]. Moreover, doxycycline exposure among carriers of *Staphylococcus aureus* and other commensal organisms has been reported to select for tetracycline resistance[12–14]. While these considerations suggest that doxyPEP implementation strategies should prioritize populations for whom the benefits of STI reduction outweigh potential risks associated with increased antibiotic consumption, evidence to inform such benefit-risk assessments is limited. Persons enrolled in randomized trials of doxyPEP[15], as well as observational studies recruiting patients in STI clinics[16], may not represent all populations who receive doxyPEP under real-world circumstances. Early cohorts of doxyPEP adopters include individuals without recent STI history, who may be at low risk of infection[17–20]. We aimed to quantify the potential impacts of doxyPEP implementation using data from nationwide commercial insurance claims. We estimated achievable reductions in incidence of STI diagnoses and associated increases in tetracycline use under current US recommendations[4], and under differing doxyPEP implementation strategies.

## Results

### Study population
We analyzed Merative™ MarketScan® Research Databases, a US-wide collection of administrative and commercial healthcare data from adjudicated claims across multiple commercial insurance plans. Our analysis period encompassed 1 January 2017 and 31 December 2019 to mitigate ithe mpacts of transient behavioral changes associated with emergency phases of the COVID-19 pandemic and subsequent mpox epidemic. We used prior-year healthcare utilization (measured from 2016 to 2018) to identify individuals likely to be prioritized for doxyPEP implementation, including PLWH, recipients of HIV pre-exposure prophylaxis (PrEP), and those with prior STI diagnoses. We restricted analyses to males and to individuals of any sex with diagnosis or procedure codes indicating receipt of gender-affirming care (see "Methods"). Whereas persons already receiving PrEP and PLWH may not encompass all individuals who could benefit from doxyPEP, we considered these populations to represent those most likely to adopt doxyPEP given their existing linkage to preventive sexual health services. Although MSM and non-MSM cannot be readily distinguished from administrative claims data, MSM comprise the vast majority of US males using PrEP[21], and account for 87% of US males living with HIV[22]. From 2017 to 2019, cohorts of PrEP recipients and PLWH eligible for analyses increased in size from 10,679 to 15,384 individuals and from 8095 to 11,011 individuals, respectively (Table S1); in total, follow-up encompassed 38,543 person-years at risk among PrEP recipients, 29,228 person-years among PLWH, and 19,918 person-years among people with prior-year STI diagnoses. Within these cohorts, between 91.4 and 93.5% of PrEP recipients and PLWH received no STI diagnoses in the preceding year; 1.1–1.9% received ≥2 STI diagnoses.

### Baseline rates
Incidence rates of gonorrhea, chlamydia, and syphilis diagnoses totaled 10.8 per 100 person-years among all PrEP recipients and 8.1 per 100 person-years among all PLWH, respectively (Table 1; Tables S2–S5). Restricting to PrEP recipients and PLWH who received ≥1 STI diagnosis in the preceding year, incidence rates of diagnoses for the same three STIs totaled 33.3–35.5 per 100 person-years. Among all MSM and PLWH, STI-associated antibiotic fill-days per 100 person-years totaled 2.2–4.5 for cephalosporins, 1.6–3.5 for macrolides, 3.3–6.2 for penicillins, and 1.1–1.4 for tetracyclines (Table 2). Incidence rates of STI-related antibiotic fill-days were roughly 3-fold higher among PrEP recipients and PLWH who received ≥1 STI diagnosis in the preceding year than among all PrEP recipients and PLWH. Overall rates of STI-related antibiotic use were highest among individuals aged 25–34 years, but similar across age groups within each risk stratum (Tables S6–S9). Most STI-related antibiotic fill-days were associated with gonorrhea infections (Table S10–S12).

### Anticipated reduction in STI burden due to doxyPEP
We modeled potential direct effects of doxyPEP on STI incidence within each population stratum, using pathogen-specific relative-risk estimates from a recent meta-analysis of randomized controlled trials of doxyPEP to quantify effect sizes[23]. We estimated that doxyPEP would prevent 2.7, 2.5, and 1.4 gonorrhea, chlamydia, and syphilis diagnoses, respectively, per 100 person-years of use among PrEP recipients (Fig. 1). Subsetting analyses to risk periods <3 months after a PrEP prescription fill or after a year of consistent PrEP use, doxyPEP would prevent 3.1–3.3 gonorrhea diagnoses, 2.9–3.0 chlamydia diagnoses, and 1.5–1.6 syphilis diagnoses per 100 person-years of use (Table S13). Considering only PrEP recipients who experienced ≥1 STI diagnosis in the preceding year, avertible incidence totaled 9.6, 8.1, and 3.1 gonorrhea, chlamydia, and syphilis diagnoses, respectively, per 100 person-years of doxyPEP use. The avertible burden comprised 1.5–2.3 diagnoses of gonorrhea, chlamydia, and syphilis per 100 person-years within the general sample of PLWH, while among PLWH with ≥1 STI diagnosis in the preceding year, rates totaled 5.9–7.4 diagnoses per 100 person-years.

### Anticipated increase in antibiotic use due to doxyPEP
We quantified changes in the use of tetracycline and other antibiotics in fill-days, defined as a day with ≥1 oral antibiotic fill or antibiotic injection. To harmonize comparisons of doxycycline consumption attributable to doxyPEP with estimated fill-days for acute STIs, we equated each prophylactic use of 200 mg doxycycline to one-seventh of an STI-associated tetracycline antibiotic fill-day. This standardization reflected the typical chlamydia treatment regimen of two 100 mg doxycycline doses daily for 7 days. We estimated that PrEP recipients and PLWH would receive the equivalent of 203.5 and 141.0 additional tetracycline fill-days, respectively, per 100 person-years of doxyPEP use (Table 3), representing 128–145 fold increases over prevailing rates of STI-related tetracycline fill-days within these populations. Among PrEP recipients and PLWH with ≥1 STI diagnosis in the preceding year, corresponding increases were equivalent to 312.9 and 271.9 additional tetracycline fills, respectively, per 100 person-years of doxyPEP use, representing 63–76 fold increases over prevailing rates of STI-related tetracycline fills within these strata. Over 100 person-years of doxyPEP use, PrEP recipients and PLWH would experience 0.8–4.8 fewer STI-related fill-days or administrations for cephalosporins, macrolides, and penicillins, while PrEP recipients and PLWH who received ≥1 STI diagnosis in the prior year would experience 3.1–9.6 fewer STI-related fills of the same drugs.

We obtained similar results using alternative estimates of the frequency with which individuals were expected to take doxyPEP based on modeled relationships between STI history and sexual risk

**Table 1 | Incidence rate of sexually transmitted infection diagnoses among males and transgender individuals from 2017 to 2019 in the United States**

| Cohort | | Incidence rate per 100 person-years (95% CI) | | | |
|---|---|---|---|---|---|
| | | Any STI | Gonorrhea | Chlamydia | Syphilis |
| PLWH | | | | | |
| | All | 8.1 (7.7, 8.6) | 3.3 (3.1, 3.6) | 1.8 (1.7, 2.0) | 3.0 (2.8, 3.2) |
| | ≥1 STI in year prior | 33.3 (30.0, 36.9) | 16.5 (14.5, 18.7) | 9.2 (7.6, 11.0) | 7.7 (6.6, 9.1) |
| | ≥2 STIs in year prior | 61.2 (51.3, 72.8) | 32.0 (26.0, 39.3) | 18.7 (14.3, 24.6) | 10.5 (7.8, 14.0) |
| | ≥3 STIs in year prior | 93.5 (71.3, 123.4) | 50.5 (37.2, 68.6) | 29.5 (18.5, 47.5) | 13.7 (8.3, 22.5) |
| PrEP use[a] | | | | | |
| | All | 10.8 (10.3, 11.2) | 5.9 (5.6, 6.2) | 3.1 (2.8, 3.3) | 1.8 (1.7, 2.0) |
| | ≥1 STI in year prior | 35.5 (32.9, 38.5) | 21.4 (19.5, 23.5) | 10.0 (8.8, 11.5) | 4.1 (3.4, 4.8) |
| | ≥2 STIs in year prior | 59.1 (51.7, 68.0) | 37.3 (31.9, 43.4) | 18.0 (14.6, 22.5) | 4.0 (2.8, 5.7) |
| | ≥3 STIs in year prior | 82.6 (64.3, 106.2) | 49.8 (37.4, 66.2) | 29.2 (20.2, 42.2) | 4.0 (1.8, 8.6) |
| Active PrEP use[b] | | | | | |
| | All | 11.5 (11.0, 12.0) | 6.4 (6.1, 6.8) | 3.3 (3.0, 3.5) | 1.8 (1.7, 2.0) |
| | ≥1 STI in year prior | 36.9 (34.1, 39.9) | 22.1 (20.2, 24.3) | 10.8 (9.6, 12.2) | 3.9 (3.3, 4.6) |
| | ≥2 STIs in year prior | 59.6 (52.5, 68.0) | 36.3 (31.0, 42.5) | 18.4 (15.3, 22.3) | 4.9 (3.7, 6.5) |
| | ≥3 STIs in year prior | 82.8 (65.5, 104.5) | 52.4 (39.7, 69.1) | 24.6 (17.7, 34.5) | 5.6 (3.3, 9.5) |
| Consistent PrEP use[c] | | | | | |
| | All | 13.0 (12.4, 13.6) | 7.3 (6.9, 7.7) | 3.7 (3.4, 4.0) | 2.1 (1.9, 2.3) |
| | ≥1 STI in year prior | 43.3 (40.0, 46.9) | 25.6 (23.3, 28.3) | 13.7 (11.9, 15.6) | 4.0 (3.2, 4.8) |
| | ≥2 STIs in year prior | 69.9 (61.2, 80.0) | 42.2 (36.1, 49.2) | 23.6 (19.0, 29.2) | 4.2 (2.8, 6.3) |
| | ≥3 STIs in year prior | 107.9 (86.6, 133.9) | 64.6 (50.2, 83.0) | 39.8 (28.1, 55.8) | 3.2 (1.2, 8.8) |
| STI history | | | | | |
| | ≥1 STI in year prior | 17.8 (16.9, 18.6) | 8.5 (8.0, 9.1) | 7.1 (6.7, 7.7) | 2.1 (1.9, 2.3) |
| | ≥2 STIs in year prior | 39.5 (36.2, 43.2) | 19.9 (17.9, 22.1) | 16.5 (14.6, 18.7) | 3.2 (2.7, 3.9) |
| | ≥3 STIs in year prior | 82.3 (71.0, 95.9) | 39.9 (33.6, 47.6) | 36.7 (29.0, 46.3) | 5.9 (4.2, 8.4) |

Incidence rates were calculated with an intercept-only general equation model with a Poisson distribution. Yearly incidence rates are reported in Tables S2–S5.
*CI* confidence interval, *STI* sexually transmitted infection, *PrEP* HIV pre-exposure prophylaxis, *PLWH* people living with HIV.
[a]PrEP users have filled ≥1 PrEP prescription in the previous year.
[b]Active PrEP users have ≥1 PrEP prescription in the previous three months.
[c]Consistent PrEP users have ≥1 PrEP prescription in the previous three months and ≥3 PrEP fills in the previous year.

(Text S1; Table S14). Reductions in STI-related antibiotic use were comparatively modest when assuming no effect of doxyPEP on gonorrhea (Table S15).

**Number-needed-to-treat for STI prevention via doxyPEP**
We next applied our estimates of doxyPEP effects on STI diagnoses and antibiotic fill-days to quantify net changes in tetracycline fill-days per incident STI diagnosis prevented. We estimated that doxyPEP implementation would result in the equivalent of 76.0–93.9, 82.5–95.7, and 61.0–145.3 additional tetracycline fill-days for each prevented diagnosis of gonorrhea, chlamydia, and syphilis, respectively, among all PrEP recipients and PLWH (Fig. 2; Table S16). For individuals within these strata who received ≥1 STI diagnosis in the preceding year, one diagnosis of gonorrhea, chlamydia, and syphilis could be prevented with the equivalent of 32.2–36.5, 37.0–38.7, and 46.1–100.2 additional tetracycline fill-days, respectively. Increases in fill-days needed to prevent each gonorrhea and chlamydia diagnosis were only modestly lower among PrEP recipients and PLWH who received ≥2 or ≥3 STI diagnoses in the prior year.

Increases in tetracycline consumption associated with doxyPEP use among all PrEP recipients or PLWH corresponded to the equivalent of 100.1–140.8, 111.1–171.7, and 29.7–80.2 additional tetracycline fill-days per averted STI-related course of treatment with cephalosporins, macrolides, and penicillins, respectively (Table 4). Restricting doxyPEP to PrEP recipients and PLWH with ≥1 STI diagnosis in the prior year, we estimated the equivalent of 43.4–55.5, 54.1–69.0, and 16.2–43.3 additional tetracycline fill-days per averted STI-related course of treatment with the same drugs.

**Overall impact of implementation**
We also estimated potential population-level changes in STI diagnoses and antibiotic use, considering scenarios where 30%, 50%, or 70% of eligible individuals adopt doxyPEP. With 50% uptake among PrEP recipients, doxyPEP would prevent 22.0%, 38.7%, and 22.6% of all diagnoses of gonorrhea, chlamydia, and syphilis, respectively, among PrEP recipients (Table 5). Use among 50% of PrEP recipients with ≥1 prior STI would prevent 0.4, 0.3, and 0.1 diagnoses of the same conditions, representing only 5.1–6.8% of diagnoses with these conditions among all PrEP recipients. Uptake among 50% of all PrEP recipients or 50% of PrEP recipients with ≥1 STI diagnosis in the prior year would increase tetracycline use by the equivalent of 101.8 and 12.7 fills per 100 person-years among all PrEP recipients, respectively, representing 73-fold and 9.1-fold increases over prevailing rates of STI-related tetracycline fills among all PrEP recipients. Restricting doxyPEP to individuals with ≥2 or ≥3 STI diagnoses in the prior year would prevent <6% of diagnoses for each STI among all PrEP recipients, while increasing STI-related tetracycline fill-days by the equivalent of 264% and 71% all PrEP recipients.

Similarly, 50% uptake of doxyPEP among all PLWH would prevent 0.7, 0.7, and 1.2 gonorrhea, chlamydia, and syphilis diagnoses, respectively, per 100 person-years (Table 5); 50% uptake among PLWH with ≥1 STI diagnosis in the prior year would prevent 0.2

**Table 2 | Incidence rate of oral and injection antibiotic fills related to sexually transmitted infection diagnoses among males and transgender individuals from 2017 to 2019 in the United States**

| Cohort | | Incidence of antibiotic fill-days related to STIs per 100 person-years (95% CI) | | | | | |
|---|---|---|---|---|---|---|---|
| | | Cephalosporins | Macrolides | Penicillins | Tetracyclines | Quinolones | Aminoglycosides |
| PLWH | | | | | | | |
| | All | 2.2 (2.0, 2.4) | 1.6 (1.4, 1.8) | 6.2 (5.7, 6.7) | 1.1 (0.9, 1.4) | 0 (0, 0.1) | 0 (0, 0) |
| | ≥1 STI in year prior | 10.8 (9.3, 12.6) | 7.8 (6.1, 10.0) | 21.8 (18.9, 25.2) | 4.3 (2.9, 6.4) | 0.1 (0, 0.4) | 0 (0, 0) |
| | ≥2 STIs in year prior | 19.3 (14.9, 24.8) | 12.6 (8.5, 19.1) | 23.1 (17.1, 31.2) | 7.1 (3.6, 14.7) | 0 (0, 0) | 0 (0, 0) |
| | ≥3 STIs in year prior | 31.5 (21.7, 45.7) | 15.7 (8.9, 32.0) | 29.4 (16.8, 51.3) | 4.2 (1.1, 16.5) | 0 (0, 0) | 0 (0, 0) |
| PrEP use[a] | | | | | | | |
| | All | 4.5 (4.2, 4.8) | 3.5 (3.3, 3.8) | 3.3 (3.0, 3.7) | 1.4 (1.2, 1.6) | 0.1 (0, 0.1) | 0 (0, 0.1) |
| | ≥1 STI in year prior | 15.9 (14.1, 18.0) | 11.1 (9.5, 13.1) | 9.4 (8.0, 11.1) | 4.1 (3.0, 5.7) | 0.1 (0.1, 0.3) | 0 (0, 0.2) |
| | ≥2 STIs in year prior | 25.0 (19.7, 31.8) | 14.3 (10.7, 19.2) | 10.2 (7.5, 13.9) | 6.0 (3.3, 10.8) | 0.1 (0, 1) | 0 (0, 0) |
| | ≥3 STIs in year prior | 35.7 (21.8, 58.4) | 16.5 (9.2, 30.2) | 10.6 (5.8, 19.3) | 9.2 (3.7, 23.4) | 0.7 (0.1, 4.6) | 0 (0, 0) |
| Active PrEP use[b] | | | | | | | |
| | All | 5.3 (5.0, 5.6) | 4.3 (3.9, 4.6) | 3.3 (3.0, 3.7) | 1.6 (1.3, 1.9) | 0.1 (0, 0.1) | 0 (0, 0.1) |
| | ≥1 STI in year prior | 17.3 (15.2, 19.7) | 12.6 (10.7, 14.8) | 9.4 (8.0, 11.0) | 4.9 (3.5, 6.7) | 0.2 (0.1, 0.5) | 0 (0, 0.3) |
| | ≥2 STIs in year prior | 25.3 (19.7, 32.5) | 16.4 (12.3, 22.0) | 10.4 (7.4, 14.6) | 6.9 (3.7, 12.8) | 0.3 (0.1, 1.3) | 0.2 (0, 1.2) |
| | ≥3 STIs in year prior | 36.5 (21.6, 61.5) | 24.8 (15.2, 40.9) | 11.7 (6.5, 21.1) | 10.2 (4.0, 26.2) | 0 (0, 0) | 0 (0, 0) |
| Consistent PrEP use[c] | | | | | | | |
| | All | 5.5 (5.1, 5.9) | 4.5 (4.1, 4.9) | 3.6 (3.3, 4.1) | 1.7 (1.4, 2.0) | 0.1 (0.1, 0.1) | 0 (0, 0.1) |
| | ≥1 STI in year prior | 17.7 (15.6, 20.0) | 13.3 (11.3, 15.8) | 9.4 (7.9, 11.2) | 5.0 (3.6, 7.2) | 0.2 (0.1, 0.5) | 0 (0, 0.3) |
| | ≥2 STIs in year prior | 26.1 (20.5, 33.2) | 17.6 (13.1, 23.7) | 9.6 (6.6, 14.1) | 7.2 (3.7, 13.8) | 0.4 (0.1, 1.5) | 0.2 (0, 1.3) |
| | ≥3 STIs in year prior | 39.8 (23.6, 67.0) | 27.1 (16.6, 44.6) | 10.3 (5.2, 20.5) | 9.5 (3.5, 26.5) | 0 (0, 0) | 0 (0, 0) |
| STI history | | | | | | | |
| | ≥1 STI in year prior | 5.5 (5.0, 5.9) | 4.9 (4.4, 5.4) | 6.0 (5.3, 6.7) | 2.1 (1.8, 2.5) | 0.1 (0, 0.1) | 0.1 (0, 0.2) |
| | ≥2 STIs in year prior | 11.3 (9.7, 13.3) | 10.2 (8.4, 12.4) | 7.3 (6.1, 8.7) | 4.0 (2.9, 5.5) | 0.1 (0, 0.3) | 0 (0, 0.2) |
| | ≥3 STIs in year prior | 24.2 (18.2, 32.2) | 16.9 (11.4, 25.5) | 11.6 (8, 16.7) | 8.1 (4.7, 14.2) | 0.2 (0, 1.4) | 0 (0, 0) |

An antibiotic was considered STI-related if commonly used as a first- or second-line treatment (Table S11) and prescribed three days prior to or following a gonorrhea or syphilis diagnosis. To capture presumptive chlamydia treatment, chlamydia related antibiotic fills were defined as doxycycline and azithromycin fills seven days before to three days after a chlamydia diagnosis. Incidence rates were calculated using an intercept-only model Poisson regression model, estimated via generalized estimating equations to address repeat observations per individual, for antibiotics associated with gonorrhea, chlamydia mono-infection, and syphilis diagnoses (Tables S10–12).

CI confidence interval, STI sexually transmitted infection, PrEP HIV pre-exposure prophylaxis, PLWH people living with HIV.

[a]PrEP use: filled ≥1 PrEP prescription in the previous year.

[b]Active PrEP use: ≥1 PrEP prescription in the previous three months.

[c]Consistent PrEP use: ≥1 PrEP prescription in the previous three months and ≥3 PrEP fills in the previous year.

## A] Gonorrhea

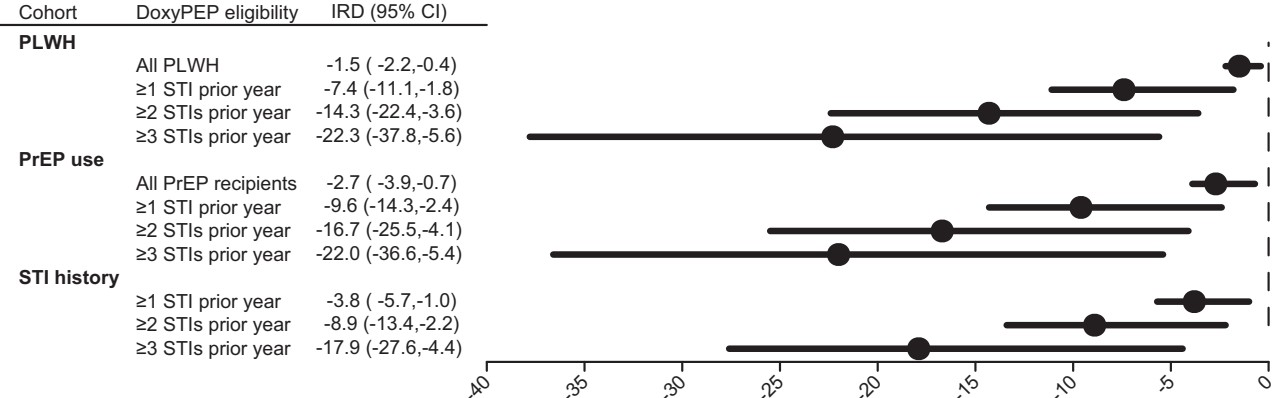

## B] Chlamydia

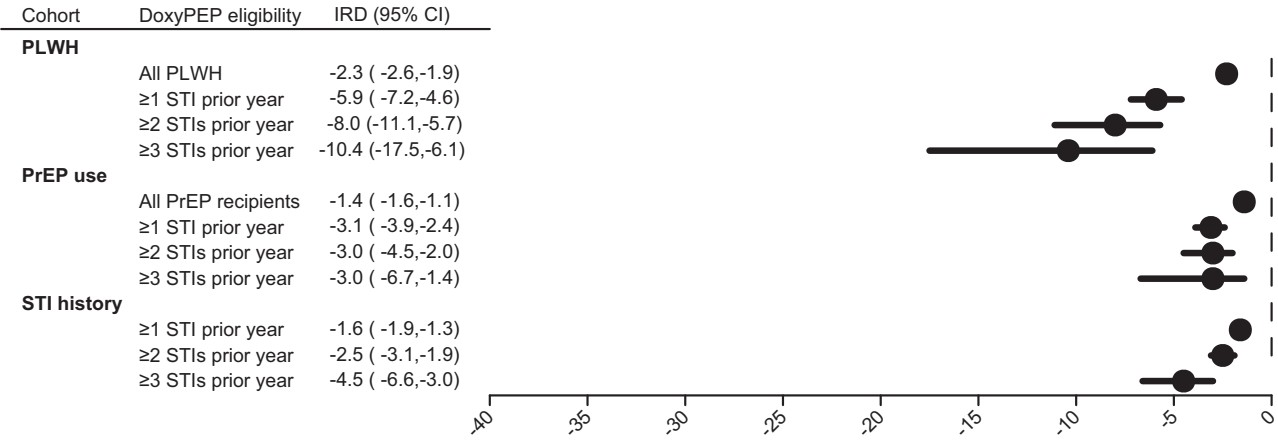

## C] Syphilis

**Incidence Rate Difference (IRD) per 100 person-years**

**Fig. 1 | Incidence of gonorrhea, chlamydia, and syphilis preventable by direct effects of doxycycline post-exposure prophylaxis (doxyPEP).** We illustrate estimates of the incidence of diagnoses of gonorrhea (**A**), chlamydia (**B**), and syphilis (**C**) preventable by direct effects of doxyPEP among recipients within differing eligibility strata. Incidence rate difference (IRD) estimates compare between rates under counterfactual scenarios of doxyPEP receipt or non-receipt. Populations included in analyses encompass males and transgender individuals living with HIV (PLWH), who receive HIV pre-exposure prophylaxis (PrEP), or with a history of diagnoses of sexually transmitted infections (STIs) in the preceding year. Follow-up for the estimates plotted encompasses 38,543 person-years at risk among PrEP recipients, 29,228 person-years at risk among PLWH, and 19,918 person-years at risk among people with prior STI diagnosis history, summed across all years of follow-up (2017–2019). Plotted points represent maximum-likelihood estimates and lines delineate 95% confidence intervals.

diagnoses of each STI per 100 person-years, representing 6.1–11.1% of such diagnoses among all PLWH. These scenarios would increase tetracycline use among PLWH by 70.5 and 9.0 fill-days per 100 person-years.

## Discussion

Using doxyPEP to prevent STIs among commercially insured individuals currently eligible to receive this intervention would require the equivalent of 32.2–36.5, 37.0–38.7, and 46.1–100.2 additional

**Table 3 | Incidence rate difference of oral and injection antibiotic fill-days related to sexually transmitted infection diagnoses among men and transgender individuals due to the use of doxycycline post-exposure prophylaxis**

| Cohort | | Incidence rate difference of antibiotic fills related to STIs per 100 person-years (95% CI) | | | |
|---|---|---|---|---|---|
| | | Cephalosporins | Macrolides | Penicillins | Tetracyclines |
| PLWH | | | | | |
| | All | −1.0 (−1.3, −0.3) | −0.8 (−1.0, −0.5) | −4.8 (−5.5, −3.8) | 141.0 (140.9, 141.2) |
| | ≥1 STI in year prior | −4.9 (−6.1, −1.4) | −3.9 (−4.2, −2.5) | −16.7 (−20.2, −13.1) | 271.9 (270.9, 272.8) |
| | ≥2 STIs in year prior | −8.7 (−9.9, −2.8) | −6.2 (−6.6, −4.8) | −17.7 (−24.4, −12.4) | 461.1 (457.3, 463.6) |
| | ≥3 STIs in year prior | −14.2 (−14.9, −5.1) | −7.4 (−8.8, −5.9) | −22.4 (−39.6, −12.5) | 647.7 (638.1, 650.1) |
| PrEP use[a] | | | | | |
| | All | −2.0 (−2.8, −0.5) | −1.8 (−2.3, −1) | −2.6 (−.03, −2.1) | 203.5 (203.4, 203.7) |
| | ≥1 STI in year prior | −7.2 (−9.3, −2.0) | −5.8 (−6.6, −3.4) | −7.2 (−8.9, −5.6) | 312.9 (312.4, 313.5) |
| | ≥2 STIs in year prior | −11.3 (−13, −3.6) | −7.6 (−8.1, −5.6) | −7.8 (−10.9, −5.5) | 421.4 (419.9, 422.8) |
| | ≥3 STIs in year prior | −15.7 (−16.2, −6.6) | −8.9 (−11.1, −6.3) | −8.1 (−14.8, −4.3) | 532.4 (526.8, 535.4) |
| Active PrEP use[b] | | | | | |
| | All | −2.4 (−3.3, −0.6) | −2.2 (−2.7, −1.2) | −2.5 (−3.0, −2.0) | 203.4 (203.3, 203.6) |
| | ≥1 STI in year prior | −7.8 (−10.1, −2.2) | −6.7 (−7.5, −4.1) | −7.2 (−8.8, −5.6) | 312.4 (311.8, 313.2) |
| | ≥2 STIs in year prior | −11.4 (−13.0, −3.7) | −8.6 (−9.2, −6.2) | −7.9 (−11.4, −5.4) | 420.7 (418.5, 422.5) |
| | ≥3 STIs in year prior | −16.0 (−16.5, −6.9) | −12.7 (−13.9, −10.0) | −8.9 (−16.2, −4.8) | 531.5 (524.8, 535.1) |
| Consistent PrEP use[c] | | | | | |
| | All | −2.5 (−3.4, −0.7) | −2.3 (−2.8, −1.2) | −2.8 (−3.3, −2.2) | 203.4 (203.2, 203.5) |
| | ≥1 STI in year prior | −8.0 (−10.3, −2.2) | −7.0 (−7.9, −4.2) | −7.2 (−8.9, −5.6) | 312.3 (311.6, 313.1) |
| | ≥2 STIs in year prior | −11.8 (−13.5, −3.7) | −9.1 (−9.7, −6.5) | −7.4 (−10.9, −4.8) | 420.6 (418.2, 422.5) |
| | ≥3 STIs in year prior | −17.4 (−18.0, −7.5) | −13.8 (−15.2, −10.9) | −7.9 (−15.7, −3.9) | 532.2 (525.0, 535.5) |
| STI history | | | | | |
| | ≥1 STI in year prior | −2.5 (−3.3, −0.7) | −2.7 (−3.1, −1.7) | −4.6 (−5.4, −3.6) | 174.6 (174.5, 174.9) |
| | ≥2 STIs in year prior | −5.1 (−6.4, −1.5) | −5.7 (−6.2, −4.1) | −5.6 (−6.9, −4.3) | 299.2 (298.7, 299.7) |
| | ≥3 STIs in year prior | −10.9 (−12.1, −3.6) | −9.4 (−10.7, −7.6) | −8.9 (−13.0, −5.9) | 447.2 (444.6, 449.2) |

*CI* confidence interval, *STI* sexually transmitted infection, *PrEP* HIV pre-exposure prophylaxis, *PLWH* people living with HIV.
[a]PrEP use: filled ≥1 PrEP prescription in the previous year.
[b]Active PrEP use: ≥1 PrEP prescription in the previous three months.
[c]Consistent PrEP use: ≥1 PrEP prescription in the previous three months and ≥3 PrEP fills in the previous year.

standardized 7-day tetracycline courses for each prevented diagnosis of gonorrhea, chlamydia, and syphilis, respectively. These increases in tetracycline consumption surpass decreases in STI-related consumption of other antibiotics by 16–69-fold margins within the prioritized populations.

Whereas current guidelines recommend doxyPEP for individuals with ≥1 STI diagnosis in the past year, early adopters of doxyPEP in several recent real-world studies have also included individuals without recent STI history. In one US study, 63% of doxyPEP recipients had not been diagnosed with any STI in the preceding year, and 15% had never been diagnosed with an STI[17]; in a second study, 52% of doxyPEP adopters had no STI diagnoses in the preceding year[18]. In studies in Italy[19] and Spain[20], 41–51%, 45–62% and 76–86% of doxyPEP recipients had never been diagnosed with gonorrhea, chlamydia, and syphilis, respectively. Prevailing doxyPEP use patterns may thus encompass populations with low baseline STI incidence. Within the broader populations of all PrEP recipients and PLWH, we estimate that the increase in tetracycline consumption resulting from doxyPEP corresponds to the equivalent of 76.0–93.9, 82.5–95.7, and 61.0–145.3 additional 7-day treatment courses for each prevented diagnosis of gonorrhea, chlamydia, and syphilis, respectively.

Restricting doxyPEP to narrower risk groups, such as those with ≥2 or ≥3 STI diagnoses in the preceding year, only modestly lowered the estimated number of tetracycline fill-days needed to prevent one diagnosis of gonorrhea or chlamydia. Even with only 50% uptake among persons with ≥2 or ≥3 STI diagnoses in the preceding year,

doxyPEP would increase tetracycline consumption by 71–264% among all PrEP recipients and PLWH in the studied populations. These limited gains in efficiency reflect the fact that individuals' recent STI history is an imperfect predictor of future risk.

Commercially insured PrEP recipients and PLWH within our cohort likely represent the populations who will receive doxyPEP under real-world conditions of implementation. However, some doxyPEP recipients may not belong to the populations studied. In two recent US studies[17,18], 28–45% of early doxyPEP adopters were HIV-negative and not currently taking PrEP at the time of doxyPEP adoption, although many had taken PrEP previously[17]. Several other real-world findings suggest that doxyPEP may reach populations who are not otherwise linked to sexual health services through PrEP or HIV care. One US study reported that 9% of males in a cohort of early doxyPEP adopters identified as heterosexual, and 14% reported no history of anal sex with male partners in the preceding 12 months[18]. Within our analyses, PrEP recipients and PLWH accounted for only 31–34% of all males who received ≥2 STI diagnoses in any preceding 1-year period.

In a previous modeling study using data from MSM and TGW who received care from a center specializing in sexual health services, doxyPEP implementation was expected to prevent 25.6–28.5 STI diagnoses per 100 person-years among PrEP recipients and PLWH[16]. Our lower estimates of preventable burden correspond to the substantially lower rates of STI diagnoses observed among PrEP recipients and PLWH in a nationwide sample of commercially insured individuals in comparison to STI clinic patients. Secondary analyses based on the

**A] Gonorrhea**

| Cohort | DoxyPEP eligibility | N (95% CI) |
|---|---|---|
| **PLWH** | | |
| | All PLWH | 93.9 (69.2,292.1) |
| | ≥1 STI prior year | 36.5 (28.4,108.0) |
| | ≥2 STIs prior year | 31.9 (27.0, 87.1) |
| | ≥3 STIs prior year | 28.4 (26.2, 70.1) |
| **PrEP use** | | |
| | All PrEP recipients | 76.0 (54.3,243.2) |
| | ≥1 STI prior year | 32.2 (24.1, 98.8) |
| | ≥2 STIs prior year | 25.0 (20.1, 72.0) |
| | ≥3 STIs prior year | 23.7 (21.6, 60.1) |
| **STI history** | | |
| | ≥1 STI prior year | 45.2 (32.7,142.9) |
| | ≥2 STIs prior year | 33.2 (25.2,100.5) |
| | ≥3 STIs prior year | 24.8 (20.2, 69.9) |

**B] Chlamydia**

| Cohort | DoxyPEP eligibility | N (95% CI) |
|---|---|---|
| **PLWH** | | |
| | All PLWH | 95.7 (80.4,141.9) |
| | ≥1 STI prior year | 37.0 (29.0, 56.2) |
| | ≥2 STIs prior year | 30.9 (22.3, 48.7) |
| | ≥3 STIs prior year | 27.8 (16.6, 49.7) |
| **PrEP use** | | |
| | All PrEP recipients | 82.5 (71.0,121.6) |
| | ≥1 STI prior year | 38.7 (31.8, 57.9) |
| | ≥2 STIs prior year | 29.2 (22.3, 44.9) |
| | ≥3 STIs prior year | 23.0 (15.2, 38.3) |
| **STI history** | | |
| | ≥1 STI prior year | 30.2 (26.0, 44.6) |
| | ≥2 STIs prior year | 22.6 (18.6, 33.6) |
| | ≥3 STIs prior year | 15.3 (11.4, 23.6) |

**C] Syphilis**

| Cohort | DoxyPEP eligibility | N (95% CI) |
|---|---|---|
| **PLWH** | | |
| | All PLWH | 61.0 ( 53.5, 75.8) |
| | ≥1 STI prior year | 46.1 ( 37.6, 59.3) |
| | ≥2 STIs prior year | 57.8 ( 41.6, 82.3) |
| | ≥3 STIs prior year | 62.2 ( 36.8,106.3) |
| **PrEP use** | | |
| | All PrEP recipients | 145.3 (126.9,180.3) |
| | ≥1 STI prior year | 100.2 ( 81.4,129.3) |
| | ≥2 STIs prior year | 139.9 ( 94.7,211.2) |
| | ≥3 STIs prior year | 176.1 ( 79.7,392.9) |
| **STI history** | | |
| | ≥1 STI prior year | 108.5 ( 93.4,135.7) |
| | ≥2 STIs prior year | 121.8 ( 97.0,159.8) |
| | ≥3 STIs prior year | 99.3 ( 68.1,148.0) |

**Excess standardized tetracycline fill-days per STI diagnosis prevented (N)**

**Fig. 2 | Number-needed-to-treat for prevention of sexually transmitted infections (STIs) via doxycycline post-exposure prophylaxis (doxyPEP).** We illustrate estimates of the excess tetracycline fill-days (standardized to resemble a 7-day treatment course for acute STI management) expected to occur for each diagnosis of gonorrhea (**A**), chlamydia (**B**), and syphilis (**C**) prevented by doxyPEP. We estimate the number-needed-to-treat by dividing the incidence rate differences for antibiotic use under counterfactual scenarios of doxyPEP receipt or non-receipt by the incidence rate difference for STI diagnoses under counterfactual scenarios of doxyPEP receipt or non-receipt. Populations encompass males and transgender individuals living with HIV (PLWH), who receive HIV pre-exposure prophylaxis (PrEP), or with a history of STI diagnoses in the preceding year. Follow-up for the estimates plotted encompasses 38,543 person-years at risk among PrEP recipients, 29,228 person-years at risk among PLWH, and 19,918 person-years at risk among people with prior STI diagnosis history, summed across all years of follow-up (2017–2019). Plotted points represent maximum-likelihood estimates and lines delineate 95% confidence intervals.

same STI clinic-based cohort have suggested that even within this study population, increases in tetracycline consumption due to doxyPEP implementation exceeded reductions in antibiotic treatment of STIs by a factor of 42[15].

In our study, gonorrhea accounted for 46% of all STI diagnoses preventable by doxyPEP among eligible PrEP recipients and 36% of preventable diagnoses among eligible PLWH. Limited doxyPEP efficacy against gonorrhea in settings with prevalent tetracycline

**Table 4 | Estimated increase in standardized tetracycline fills per STI-related antibiotic prevented by use of doxycycline post-exposure prophylaxis**

| DoxyPEP-eligible population | Cohort | Increase in standardized tetracycline fill-days per other antibiotic fill-day prevented; N (95% CI) | | |
|---|---|---|---|---|
| | | Cephalosporins | Macrolides | Penicillins |
| Entire cohort | | | | |
| | PLWH | 140.8 (105.3, 431.7) | 171.7 (146.9, 307.8) | 29.7 (25.9, 36.9) |
| | PrEP use[a] | 100.1 (72.1, 317.4) | 111.1 (90.2, 210.7) | 79.5 (68.5, 99.2) |
| | Active PrEP use[b] | 85.2 (61.5, 269.5) | 92.5 (75.3, 173.8) | 80.2 (68.7, 100.2) |
| | Consistent PrEP use[c] | 81.8 (59.4, 257.6) | 88.8 (72.6, 168.6) | 72.7 (61.8, 91.3) |
| ≥1 STI in year prior | | | | |
| | PLWH | 55.5 (44.4, 160.2) | 69.0 (64.2, 106.9) | 16.2 (13.4, 20.8) |
| | PrEP use[a] | 43.4 (33.5, 129.0) | 54.1 (47.4, 92.2) | 43.3 (35.3, 55.9) |
| | Active PrEP use[b] | 39.9 (30.9, 118.2) | 47.0 (41.6, 75.9) | 43.5 (35.5, 56.0) |
| | Consistent PrEP use[c] | 39.0 (30.2, 116.1) | 44.9 (39.8, 74.2) | 43.3 (35.0, 56.1) |
| | STI history alone | 70.5 (52.0, 218.6) | 64.8 (55.6, 103.0) | 38.2 (32.4, 48.0) |
| ≥2 STIs in year prior | | | | |
| | PLWH | 52.9 (47.0, 138.2) | 74.1 (70.2, 95.6) | 26.1 (18.8, 37.3) |
| | PrEP use[a] | 37.2 (32.5, 99.0) | 55.8 (52.2, 75.4) | 53.9 (38.7, 77.1) |
| | Active PrEP use[b] | 36.7 (32.4, 96.5) | 48.8 (45.8, 67.2) | 53.0 (36.8, 78.0) |
| | Consistent PrEP use[c] | 35.6 (31.2, 94.4) | 46.2 (43.5, 64.7) | 57.2 (38.3, 87.3) |
| | STI history alone | 58.3 (46.6, 168.2) | 52.7 (48.3, 73.8) | 53.4 (43.0, 69.4) |
| ≥3 STIs in year prior | | | | |
| | PLWH | 45.6 (43.6, 105.7) | 87.2 (72.8, 110.4) | 28.9 (16.1, 52.1) |
| | PrEP use[a] | 33.9 (32.9, 68.6) | 59.5 (47.4, 85.4) | 66.0 (35.5, 124.8) |
| | Active PrEP use[b] | 33.3 (32.2, 65.0) | 42.0 (38.0, 53.4) | 59.7 (32.4, 111.7) |
| | Consistent PrEP use[c] | 30.5 (29.5, 59.8) | 38.5 (34.9, 48.9) | 67.3 (33.4, 137.5) |
| | STI history alone | 40.8 (37.2, 103.8) | 47.6 (41.7, 59.2) | 50.5 (34.2, 76.4) |

*CI* confidence interval, *STI* sexually transmitted infection, *PrEP* HIV pre-exposure prophylaxis, *PLWH* people living with HIV, *doxyPEP* doxycycline post-exposure prophylaxis.
[a]PrEP use: filled ≥1 PrEP prescription in the previous year.
[b]Active PrEP use: ≥1 PrEP prescription in the previous three months.
[c]Consistent PrEP use: ≥1 PrEP prescription in the previous three months and ≥3 PrEP fills in the previous year.

resistance in *N. gonorrhoeae* demonstrates that resistance selection may undermine the benefits of this intervention[6,24]. Emerging data demonstrating associations of doxyPEP use with tetracycline resistance in *N. gonorrhoeae* isolates[25] complement prior modeling studies in raising concern about resistance selection under this intervention. Linked resistance patterns across multiple antibiotic classes in *N. gonorrhoeae* further suggest selective pressure driven by doxyPEP may also impact the efficacy of other antibiotics used for gonorrhea treatment[26,27].

Our study has several limitations. First, we do not consider population-level effects of doxyPEP associated with reducing STI transmission. Whereas our analysis addresses only STI diagnoses, the prevention of undiagnosed mild or asymptomatic infections may yield indirect benefit by reducing individuals' risk of STI exposure. However, a prior modeling study estimated that such benefits could be offset by increases in tetracycline resistance, with doxyPEP expected to lose clinical effectiveness against gonorrhea within 2–12 years[28]. Moreover, asymptomatic extragenital *C. trachomatis* and *N. gonorrhoeae* infections typically detected through screening have been associated with low bacterial load or viability as well as spontaneous resolution[29–31], suggesting that preventing such diagnoses through the use of doxyPEP may have a limited impact on transmission. Decreases in population-level STI transmission due to doxyPEP would reduce the individual-level clinical benefit that our study estimates is achievable through doxyPEP use, increasing the number of doxyPEP courses needed to prevent one STI episode. Second, claims data are limited in distinguishing new-onset or historical diagnoses, may suggest nonspecific linkages between antibiotics and

STI diagnoses, and may fail to identify sexual and gender-minority populations. These data do not enable restriction to MSM and TGW, although real-world use of doxyPEP has not been strictly limited to MSM and TGW[17]. Third, our analyses may not fully capture STI-related antibiotic dispenses if some antibiotics are purchased without insurance reimbursement, received via expedited partner therapy, or obtained through public health clinics or informal sources. Fourth, we assumed one doxycycline dose for every casual or first-time condomless anal sex partnership. As doxyPEP is also recommended after condomless vaginal or oral sexual exposures, estimates may reflect lower bounds for impacts on real-world tetracycline use. Rates of STI diagnoses increased over the study period[1], meaning our results may also under-estimate avertible STI burden. Continually updated epidemiologic and risk behavior data will provide an important basis for anticipating the effects of doxyPEP on both STI diagnoses and antibiotic use. Fifth, volume and frequency of antibiotic use do not directly measure selective pressure, and alternative outcomes such as antibiotic spectrum index[32], days of antibiotic spectrum coverage[33], or other metrics for antibiotic pressure[34,35] may better quantify the implications of doxyPEP-related changes in antibiotic use for resistance selection. Finally, our cohort was limited to individuals enrolled in commercial insurance plans. Prior studies have demonstrated that lack of insurance is an important predictor of non-adoption or discontinuation of PrEP among US MSM[36–38], suggesting our sample may represent the populations most likely to use doxyPEP under real-world conditions. However, high STI risk and lack of insurance among disadvantaged groups remain important considerations[39], with doxyPEP awareness

**Table 5 | Estimated overall cohort incidence rate difference of sexually transmitted infection diagnoses and standardized tetracycline fill-days among males and transgender individuals by doxycycline post-exposure prophylaxis uptake and implementation strategy**

| Uptake scenario | Cohort | DoxyPEP-eligible population | Incidence rate difference per 100 person-years (95% CI) | | | |
|---|---|---|---|---|---|---|
| | | | Gonorrhea | Chlamydia | Syphilis | Tetracycline fill-days |
| 50% uptake | | | | | | |
| | PLWH | | | | | |
| | | All | −0.7 (−1.1, −0.2) | −0.7 (−0.9, −0.5) | −1.2 (−1.3, −0.9) | 70.5 (70.5, 70.6) |
| | | ≥1 STI in year prior | −0.2 (−0.3, −0.1) | −0.2 (−0.3, −0.2) | −0.2 (−0.2, −0.2) | 9.0 (9.0, 9.0) |
| | | ≥2 STIs in year prior | −0.1 (−0.1, 0) | −0.1 (−0.1, −0.1) | 0 (−0.1, 0) | 2.9 (2.8, 2.9) |
| | | ≥3 STIs in year prior | 0 (0, 0) | 0 (−0.1, 0) | 0 (0, 0) | 1.1 (1.0, 1.1) |
| | PrEP use[a] | | | | | |
| | | All | −1.3 (−2.0, −0.3) | −1.2 (−1.4, −0.8) | −0.7 (−0.8, −0.6) | 101.8 (101.7, 101.8) |
| | | ≥1 STI in year prior | −0.4 (−0.5, −0.1) | −0.3 (−0.4, −0.2) | −0.1 (−0.2, −0.1) | 12.7 (12.7, 12.8) |
| | | ≥2 STIs in year prior | −0.1 (−0.2, 0) | −0.1 (−0.2, −0.1) | 0 (0, 0) | 3.7 (3.7, 3.8) |
| | | ≥3 STIs in year prior | 0 (0, 0) | 0 (−0.1, 0) | 0 (0, 0) | 1.1 (1.0, 1.1) |
| | STI history | | | | | |
| | | ≥1 STI in year prior | −1.9 (−2.8, −0.5) | −2.9 (−3.3, −2.0) | −0.8 (−0.9, −0.6) | 87.3 (87.2, 87.4) |
| | | ≥2 STIs in year prior | −0.7 (−0.9, −0.2) | −1 (−1.3, −0.7) | −0.2 (−0.2, −0.1) | 23.4 (23.4, 23.4) |
| | | ≥3 STIs in year prior | −0.2 (−0.3, −0.1) | −0.4 (−0.5, −0.2) | −0.1 (−0.1, 0) | 5.5 (5.5, 5.5) |
| 30% uptake | | | | | | |
| | PLWH | | | | | |
| | | All | −0.4 (−0.7, −0.1) | −0.4 (−0.5, −0.3) | −0.7 (−0.8, −0.6) | 42.3 (42.3, 42.4) |
| | | ≥1 STI in year prior | −0.1 (−0.2, 0.0) | −0.1 (−0.2, −0.1) | −0.1 (−0.1, −0.1) | 5.4 (5.4, 5.4) |
| | | ≥2 STIs in year prior | 0.0 (0.0, 0.0) | 0.0 (−0.1, 0.0) | 0.0 (0.0, 0.0) | 1.7 (1.7, 1.7) |
| | | ≥3 STIs in year prior | 0.0 (0.0, 0.0) | 0.0 (0.0, 0.0) | 0.0 (0.0, 0.0) | 0.6 (0.6, 0.6) |
| | PrEP use[a] | | | | | |
| | | All | −0.8 (−1.2, −0.2) | −0.7 (−0.9, −0.5) | −0.4 (−0.5, −0.3) | 61.1 (61.0, 61.1) |
| | | ≥1 STI in year prior | −0.3 (−0.3, −0.1) | −0.2 (−0.3, −0.1) | −0.1 (−0.1, 0.0) | 7.6 (7.6, 7.7) |
| | | ≥2 STIs in year prior | −0.1 (−0.1, 0.0) | −0.1 (−0.1, 0.0) | 0.0 (0.0, 0.0) | 2.2 (2.2, 2.2) |
| | | ≥3 STIs in year prior | 0.0 (0.0, 0.0) | 0.0 (0.0, 0.0) | 0.0 (0.0, 0.0) | 0.6 (0.6, 0.6) |
| | STI history | | | | | |
| | | ≥1 STI in year prior | −1.2 (−1.7, −0.3) | −1.7 (−2, −1.2) | −0.5 (−0.6, −0.4) | 52.4 (52.3, 52.5) |
| | | ≥2 STIs in year prior | −0.4 (−0.6, −0.1) | −0.6 (−0.8, −0.4) | −0.1 (−0.1, −0.1) | 14.1 (14.0, 14.1) |
| | | ≥3 STIs in year prior | −0.1 (−0.2, 0.0) | −0.2 (−0.3, −0.1) | 0.0 (0.0, 0.0) | 3.3 (3.3, 3.3) |
| 70% uptake | | | | | | |
| | PLWH | | | | | |
| | | All | −1.0 (−1.5, −0.3) | −1.0 (−1.2, −0.7) | −1.6 (−1.8, −1.3) | 98.7 (98.6, 98.8) |
| | | ≥1 STI in year prior | −0.3 (−0.4, −0.1) | −0.3 (−0.4, −0.2) | −0.3 (−0.3, −0.2) | 12.6 (12.5, 12.6) |
| | | ≥2 STIs in year prior | −0.1 (−0.1, 0) | −0.1 (−0.2, −0.1) | −0.1 (−0.1, 0) | 4.0 (4.0, 4.0) |
| | | ≥3 STIs in year prior | −0.1 (−0.1, 0) | −0.1 (−0.1, 0) | 0 (0, 0) | 1.5 (1.5, 1.5) |
| | PrEP use[a] | | | | | |
| | | All | −1.9 (−2.7, −0.5) | −1.7 (−2, −1.2) | −1.0 (−1.1, −0.8) | 142.5 (142.4, 142.6) |
| | | ≥1 STI in year prior | −0.6 (−0.7, −0.2) | −0.5 (−0.6, −0.3) | −0.2 (−0.2, −0.1) | 17.8 (17.8, 17.9) |
| | | ≥2 STIs in year prior | −0.2 (−0.3, −0.1) | −0.2 (−0.2, −0.1) | 0 (−0.1, 0) | 5.2 (5.2, 5.2) |
| | | ≥3 STIs in year prior | −0.1 (−0.1, 0) | −0.1 (−0.1, 0) | 0 (0, 0) | 1.5 (1.4, 1.5) |
| | STI history | | | | | |
| | | ≥1 STI in year prior | −2.7 (−4.0, −0.7) | −4.0 (−4.7, −2.7) | −1.1 (−1.3, −0.9) | 122.2 (122.1, 122.4) |
| | | ≥2 STIs in year prior | −1.0 (−1.3, −0.3) | −1.5 (−1.8, −1) | −0.3 (−0.3, −0.2) | 32.8 (32.7, 32.8) |
| | | ≥3 STIs in year prior | −0.3 (−0.4, −0.1) | −0.5 (−0.7, −0.3) | −0.1 (−0.1, −0.1) | 7.7 (7.7, 7.8) |

Incidence rate difference estimates can be interpreted as the cohort-wide change in STI-related antibiotic fills within each cohort for each uptake scenario, with doxyPEP targeted to the indicated eligible population. The STI history cohort contains all individuals with at least one STI in the prior year.

*CI* confidence interval, *STI* sexually transmitted infection, *PrEP* HIV pre-exposure prophylaxis, *PLWH* people living with HIV, *doxyPEP* doxycycline post-exposure prophylaxis.

[a]PrEP use: filled ≥1 PrEP prescription in the previous year.

currently low among rural MSM and those without postsecondary education[40]. Expanding awareness of doxyPEP among providers, implementing this intervention in public STI clinics, and pursuing flexible financing mechanisms for this intervention for both insured and uninsured individuals may be important strategies to achieve equitable access.

Our findings offer a quantitative basis for evaluating potential benefit-risk tradeoffs of differing doxyPEP implementation strategies.

Benefits of doxyPEP alongside STI prevention may include alleviation of the medical costs, productivity losses, and stigma associated with STI diagnoses. However, preliminary data have identified increased prevalence of tetracycline resistance in *S. aureus*, group A *Streptococcus*, and the gut microbiome among doxyPEP recipients[7,14,25], and in *S. aureus* infections within populations eligible for doxyPEP use[12]. Ongoing surveillance for antimicrobial resistance in *N. gonorrhoeae* and other bacterial pathogens[41] remains a priority to understand implications for clinical care.

## Methods

### Study design, cohorts, and clinical outcomes

The study population for this retrospective cohort study using the Merative™ MarketScan® Research Databases consisted of commercially insured males and transgender individuals (Table S17) aged 18–64 years at any time between 1 January 2017 and 31 December 2019. We included time-at-risk for each individual following any 12-month period (extending as early as 1 January, 2016) during which they were continuously enrolled in both medical and prescription coverage (no lapse >60 days) and filled ≥1 prescription.

We used International Classification of Diseases, Tenth Revision (ICD-10) codes and National Drug Codes (NDCs) to identify individuals likely belonging to population strata prioritized for doxyPEP implementation (Tables S18–S22). Our primary analyses encompassed persons receiving PrEP and PLWH. To identify PrEP recipients, we used a previously validated algorithm[42] identifying individuals who filled prescriptions for tenofovir disoproxil and emtricitabine (TDF-FTC), excluding those with a needlestick exposure within ≤10 days of the prescription fill or any prior hepatitis B, HIV, or AIDS-defining opportunistic infection diagnoses.

We also defined strata of individuals who had received ≥1, ≥2, or ≥3 diagnoses of chlamydia, gonorrhea, or syphilis in the preceding 12 months (as described below; Tables S23–S25). While both MSM and non-MSM could be represented among persons with prior bacterial STI diagnoses, MSM are over-represented within this population[43], and the clinical rationale for doxyPEP applies to all individuals at high risk of bacterial STIs, regardless of the gender of their sexual partners.

As PrEP use may be intermittent, we further sought to distinguish person-time at risk associated with periods during which individuals were expected to be using PrEP actively. Among all individuals who had received ≥1 PrEP fill, we distinguished follow-up occurring within 3 months after a new prescription fill as periods with "active" PrEP prescriptions, reflecting the typical 3-month (daily use) supply per dispense. We also distinguished risk periods where individuals had received ≥3 PrEP fills within the preceding 12 months as periods where we expected individuals were engaged in "consistent" PrEP use. Claims data do not distinguish between daily and event-driven (i.e., "2-1-1") patterns of PrEP use. Both daily and event-driven use can be represented in our primary analyses (encompassing all PrEP recipients) and in the "active" use period (≤3 months after any fill). However, event-driven use may be under-represented in the "consistent" use pattern if such use does not necessitate ≥3 prescription fills in any 365-day period. For our primary analyses addressing any PrEP use, misclassification of event-driven PrEP use as non-use would occur only for individuals receiving 0 PrEP prescription fills in the preceding 12 months.

Study outcomes were diagnoses and class-specific antibiotic fill-days for bacterial STIs (chlamydia, gonorrhea, and syphilis infections). We limited gonorrhea and chlamydia diagnoses to those occurring ≥30 days after any prior diagnosis. We limited syphilis diagnoses to those occurring within 10 days of a penicillin injection and ≥365 days after any prior syphilis diagnosis to mitigate the risk of misclassifying historical or duplicate infections as new diagnoses. Chlamydia mono-infection diagnoses were those occurring without accompanying gonorrhea diagnoses within periods 3 days before to 7 days after the chlamydia diagnosis. We identified oral antibiotic prescriptions using NDCs with therapeutic classifications of 4 or 6–20, and identified injectable antibiotics via Healthcare Common Procedure Coding System (HCPCS) codes (Table S26). An antibiotic fill-day was defined as a day with ≥1 oral antibiotic fill or antibiotic injection. As syphilis treatment strategies frequently involve injections over multiple days, multiple fill-days could result from a single syphilis episode. We defined STI-associated fills with first- or second-line antibiotics (Table S27) as those occurring within 3 days before or after syphilis and gonorrhea diagnoses, and within 7 days before to 3 days after a chlamydia diagnosis (allowing a long period to capture antibiotics received prior to chlamydia diagnoses to account for presumptive treatment during gonorrhea co-infection).

### Descriptive analyses and baseline rates

We present descriptive characteristics of the study population as ranges of rates or proportions across cohorts eligible for analysis in 2016, 2017, and 2018 (Table S1). We calculated incidence rates of chlamydia, gonorrhea, and syphilis diagnoses, and of antibiotic fill-days associated with these outcomes. We defined an aggregate incidence rate for STI-related antibiotic fill-days associated with ≥1 syphilis, gonorrhea, or chlamydia diagnosis to avoid double-counting of overlapping treatment regimens targeting chlamydia/gonorrhea co-infection. We stratified rates for PrEP recipients, PLWH, and individuals with prior STI diagnoses, and computed subgroup-specific rates within each cohort according to individuals' number of STI diagnoses in the preceding 12 months. We obtained incidence rates and accompanying 95% confidence intervals via generalized estimating equations with a Poisson link function.

### Modeling analyses

Within each population stratum, we estimated the potential direct effect of doxyPEP on the incidence rate of each outcome. We parameterized direct effects of doxyPEP in preventing each infection, and first-line STI-related antibiotic fill-days for the same infection, using pathogen-specific relative-risk estimates from a recent meta-analysis of randomized controlled trials of doxyPEP[23]. We did not consider treatment effects on STI-related antibiotic fill-days for second-line antibiotic classes; *N. gonorrhoeae* infections resistant to cephalosporins and second-line therapies are also typically resistant to tetracycline and thus may not be prevented by doxyPEP[44]. As one doxyPEP trial[6] did not report statistically significant effects of doxyPEP on gonorrhea infections, we conducted sensitivity analyses including effects on chlamydia and syphilis only.

To model possible increases in standardized tetracycline fill-days expected to result from doxyPEP use, we estimated the rate of casual or first-time condomless anal sex partnerships among populations prioritized for doxyPEP implementation using data from a previous study of sexual encounter frequencies and characteristics among MSM. These data-defined rates for MSM generally, and for MSM using PrEP and MSM with HIV[45,46]. To account for expected differences in the frequency of sexual encounters across population strata with differing STI history, we used data from three studies[47–49] measuring associations between MSM's risk of bacterial STI diagnoses and their number of recent sex partners (Table S28; Text S1).

We quantified increases in antibiotic consumption due to doxyPEP implementation by adding one standardized fill-day for every 7 anticipated prophylactic uses of doxycycline after condomless anal sex with new or casual partners. We estimated incidence rate differences (IRDs) for tetracycline fill-days accounting for additional fill-days due to doxyPEP, less STI-associated tetracycline fill-days prevented.

We also estimated potential population-level changes in incidence of STI diagnoses and STI-related antibiotic use, assuming varying levels

of doxyPEP uptake within each recipient stratum. We computed IRDs for diagnoses of gonorrhea, chlamydia, and syphilis as well as IRDs for fill-days for each antibiotic class, considering scenarios where doxyPEP is available for individuals with ≥1, ≥2, or ≥3 STI diagnoses in the prior year and where 30%, 50%, or 70% of eligible individuals adopt doxyPEP.

Last, to facilitate interpretation, we used our IRD estimates for STI diagnoses and antibiotic fill-days to quantify net changes in tetracycline consumption per STI diagnosis prevented by doxyPEP. This estimate can be interpreted as a "number needed to treat"[50] for doxyPEP, defined as the net increase in standardized tetracycline fill-days per incident STI diagnosis prevented. We obtained this value by dividing IRDs for tetracycline fill-days by IRDs for gonorrhea, chlamydia, and syphilis diagnoses.

## Software
We conducted analyses using R software (R Foundation for Statistical Computing, Vienna, Austria).

## Ethics
These analyses of deidentified insurance claims data were considered exempt from review by the University of Alabama at Birmingham Institutional Review Board and the Committee for the Protection of Human Subjects at the University of California, Berkeley.

## Reporting summary
Further information on research design is available in the Nature Portfolio Reporting Summary linked to this article.

## Data availability
Raw data from the MarketScan insurance claims databases are available for licensed users. A user license could be obtained by following the instructions at https://marketscan.truvenhealth.com/marketscanportal/.

## Code availability
Analysis code is available from GitHub[51].

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

## Acknowledgements

This work was supported by intramural funding from the University of Alabama at Birmingham (K.J.B.) and the University of California, Berkeley (A.M.P.).

## Author contributions

A.M.P., J.A.L., and K.J.B. contributed to the study concept and design. A.M.P. and L.C. led the acquisition of the data. A.M.P. and L.M.K. led statistical analysis of data. A.M.P., J.J.C., S.I.M., J.A.L., and K.J.B. led the interpretation of the data. A.M.P., J.A.L., and K.J.B. drafted the manuscript, and all authors critically revised the manuscript for important intellectual content. K.J.B. obtained funding. J.A.L. and K.J.B. provided supervision.

## Competing interests

The authors declare no competing interests.

## Additional information

[1]Division of Epidemiology, School of Public Health, University of California, Berkeley, Berkeley, CA, USA. [2]Department of Infectious Diseases, Los Angeles Medical Center, Southern California Permanente Medical Group, Los Angeles, CA, USA. [3]Perisphere Real World Evidence, Austin, TX, USA. [4]Gilead Sciences, Inc., Foster City, CA, USA. [5]Division of Infectious Diseases & Vaccinology, School of Public Health, University of California, Berkeley, Berkeley, CA, USA. [6]Center for Computational Biology, College of Computing, Data Science, and Society, University of California, Berkeley, Berkeley, CA, USA. [7]Department of Epidemiology, University of Alabama at Birmingham, Birmingham, AL, USA. [8]These authors contributed equally: Joseph A. Lewnard, Katia J. Bruxvoort. ✉e-mail: jLewnard@berkeley.edu

