## [Transparent Peer Review file · Nature Communications]

Bacterial sexually transmitted infections and related antibiotic use among individuals eligible for doxycycline post-exposure prophylaxis in the United States

Corresponding Author: Dr Joseph Lewnard

Version 0:

Reviewer comments:

Reviewer #1

(Remarks to the Author)

Thank you for the opportunity to review the manuscript, “Bacterial sexually transmitted infections and related antibiotic use among individuals eligible for doxycycline postexposure prophylaxis in the United States.” In brief, this manuscript uses national commercial insurance claims data from the Merative MarketScan® databases (2016–2019) to estimate STI incidence and antibiotic use among U.S. males and transgender individuals eligible for doxycycline postexposure prophylaxis (doxyPEP). The authors model potential impacts of implementing doxyPEP among various subpopulations (e.g., PrEP users, people living with HIV [PLWH], and individuals with prior-year STI diagnoses). Key findings include high STI incidence among individuals with prior STI history, substantial increases in tetracycline use with doxyPEP, and modest declines in STI incidence. The study provides a quantitative estimate of number-needed-to-treat and antibiotic consumption trade-offs to inform implementation strategies.

This is a timely and interesting manuscript. While it parallels prior work (e.g., Traeger et al.), it draws on a broader and different dataset, offering unique insights. I have several questions about the methods and modeling assumptions that, if addressed, would strengthen the manuscript’s contribution.

Specific comments by section:

Abstract:

- The abstract clearly summarizes the study’s purpose, methods, and key findings.

Introduction:

- The sentence “We therefore aimed to quantify potential impacts of doxyPEP implementation under current US recommendations...” implies the analysis is limited to populations explicitly covered by the 2024 CDC guidelines (i.e., MSM and TGW with a prior-year STI). However, the modeling clearly extends beyond this, including broader uptake among all PrEP users, PLWH, and individuals with no recent STI history. This broader scope is important especially given real-world uptake patterns and should be explicitly acknowledged earlier. I recommend clarifying in the Introduction that the analysis includes both guideline-aligned and off-guideline scenarios.

Methods:

- The use of a large, national claims dataset is a strength and provides generalizability to commercially insured populations. Definitions for cohort stratification (e.g., PrEP, PLWH) are clearly described in the supplementary material.
- The method of equating each doxyPEP use to 1/7 of a doxycycline treatment course is a clever standardization and works well for chlamydia, where doxycycline is the preferred treatment. However, this comparison is less informative for gonorrhea and syphilis, which are typically treated with ceftriaxone and penicillin, respectively—agents with broader spectrum and different resistance implications. I recommend acknowledging this limitation and considering whether a complementary metric (e.g., spectrum index or “antibiotic pressure” score) might better capture comparative stewardship impact.
- The reliance on ICD-10 codes and pharmacy claims to define STI diagnoses introduces limitations, particularly for syphilis, where codes may not distinguish between active vs. historical infections or disease stages. Some coding choices also raise concerns: for example, inclusion of IV azithromycin 500mg—rarely used in sexual health—could inflate STI-related macrolide estimates. Likewise, the use of broad or non-anatomically specific chlamydia codes (e.g., “other chlamydial

diseases") may misclassify non-STI infections. Please review and refine the code lists where appropriate, and clearly acknowledge these limitations.

- The analysis does not appear to account for on-demand (event-driven) PrEP use, which is increasingly common. By equating "active" or "consistent" PrEP use with ongoing sexual risk and likely doxyPEP candidacy, the model may overestimate doxyPEP frequency among those with intermittent risk. Please acknowledge this in the manuscript and, if feasible, discuss implications for the model's projections.

- The assumption that doxyPEP uptake will mirror PrEP use—both in terms of population and adherence—is not explicitly justified. While PrEP users are likely early adopters, real-world data suggest some doxyPEP users are not on PrEP or have different behavioral profiles. I recommend acknowledging this assumption and supporting it with relevant citations or qualifying its limitations in the modeling framework.

Results

• Figures 1 and 2 (pp. 15–16) clearly and effectively illustrate the modeled trade-offs between STI prevention and antibiotic use across subgroups.

Discussion

• Please clarify the implications of declining doxyPEP effectiveness for gonorrhea over time due to emerging tetracycline resistance. While Table S15 provides a sensitivity analysis assuming no effect on gonorrhea, this important scenario deserves more discussion, particularly given that gonorrhea reduction appears to be the largest driver of benefit.

• The study period (2016–2019) overlaps with a time of increasing PrEP use, declining condom use, and rising STI rates. These secular trends suggest that modeled estimates may underestimate today's STI incidence and avertible burden. The authors should acknowledge that their findings may be conservative relative to current dynamics and that newer behavioral data could further inform implementation projections.

• The manuscript notes disparities in doxyPEP awareness and use (e.g., among uninsured individuals, rural MSM), but stops short of exploring how the findings might inform equitable implementation. I encourage the authors to expand on how their results could guide more equitable policy design, such as integrating doxyPEP into public STI clinics or ensuring access among underserved populations.

Reviewer #2

(Remarks to the Author)

The analysis is timely and adds important context for the discussion on benefits and disadvantages of doxyPEP

- Abstract: please include information on the type of model used
- Observed diagnosis rate is not the same as unobserved incidence rate, and referring to the outcome of interest consistently as diagnosis rates would improve clarity
- This above difference is significant also for interpretation of the results. Not all incident infections are diagnosed, and key estimates in this study quantify increases in antibiotic use compared to diagnosed and treated infections. The analysis does not capture the number of infections averted, in people using doxyPEP, that would not have been diagnosed. While this would not change the conclusions of the study when using diagnosis rate as the unit of measure, I would encourage you to discuss the full range of potential direct benefits of doxyPEP, and limitations of the data in capturing these.
- A brief explanation for conversion rates for different antibiotic modalities (treatment vs doxyPEP) is helpful; consider placing this in the main-text for clarity.
- The abbreviation IRD is defined in a figure, but not in the method text (methods line 311).

Version 1:

Reviewer comments:

Reviewer #1

(Remarks to the Author)

Thank you for the opportunity to re-review this manuscript. I appreciate the authors' efforts to address the prior comments. I am satisfied with the revisions and have no additional comments.

(Remarks on code availability)

REVIEWER COMMENTS

Reviewer #1 (Remarks to the Author):

Thank you for the opportunity to review the manuscript, “Bacterial sexually transmitted infections and related antibiotic use among individuals eligible for doxycycline postexposure prophylaxis in the United States.” In brief, this manuscript uses national commercial insurance claims data from the Merative MarketScan® databases (2016–2019) to estimate STI incidence and antibiotic use among U.S. males and transgender individuals eligible for doxycycline postexposure prophylaxis (doxyPEP). The authors model potential impacts of implementing doxyPEP among various subpopulations (e.g., PrEP users, people living with HIV [PLWH], and individuals with prior-year STI diagnoses). Key findings include high STI incidence among individuals with prior STI history, substantial increases in tetracycline use with doxyPEP, and modest declines in STI incidence. The study provides a quantitative estimate of number-needed-to-treat and antibiotic consumption trade-offs to inform implementation strategies.

This is a timely and interesting manuscript. While it parallels prior work (e.g., Traeger et al.), it draws on a broader and different dataset, offering unique insights. I have several questions about the methods and modeling assumptions that, if addressed, would strengthen the manuscript’s contribution.

We thank the Reviewer for this assessment and for the helpful comments below, each of which we have addressed in this revision. We are confident these edits have strengthened the manuscript.

Specific comments by section:

Abstract:

- The abstract clearly summarizes the study’s purpose, methods, and key findings.

We thank the Reviewer for this assessment.

Introduction:

- The sentence “We therefore aimed to quantify potential impacts of doxyPEP implementation under current US recommendations...” implies the analysis is limited to populations explicitly covered by the 2024 CDC guidelines (i.e., MSM and TGW with a prior-year STI). However, the modeling clearly extends beyond this, including broader uptake among all PrEP users, PLWH, and individuals with no recent STI history. This broader scope is important especially given real-world uptake patterns and should be explicitly acknowledged earlier. I recommend clarifying in the Introduction that the analysis includes both guideline-aligned and off-guideline scenarios.

We thank the Reviewer for this suggestion and have revised the Introduction to mention both the evidence of emerging “off-guideline” use and the fact that our analysis addresses this (lines 73-74 and 75-77).

Methods:

- The use of a large, national claims dataset is a strength and provides generalizability to commercially insured populations. Definitions for cohort stratification (e.g., PrEP, PLWH) are clearly described in the supplementary material.

We thank the Reviewer for this assessment.

- The method of equating each doxyPEP use to 1/7 of a doxycycline treatment course is a clever standardization and works well for chlamydia, where doxycycline is the preferred treatment. However, this comparison is less informative for gonorrhea and syphilis, which are typically treated with ceftriaxone and penicillin, respectively—agents with broader spectrum and different resistance implications. I recommend acknowledging this limitation and considering whether a complementary metric (e.g., spectrum index or “antibiotic pressure” score) might better capture comparative stewardship impact.

We agree with this point and have revised the Discussion (limitations paragraph) to address that these antibiotic classes have distinct implications for resistance selection. We have cited recent studies proposing alternative metrics such as antibiotic spectrum index, pressure score, and days of spectrum coverage (lines 237-241).

- The reliance on ICD-10 codes and pharmacy claims to define STI diagnoses introduces limitations, particularly for syphilis, where codes may not distinguish between active vs. historical infections or disease stages. Some coding choices also raise concerns: for example, inclusion of IV azithromycin 500mg—rarely used in sexual health—could inflate STI-related macrolide estimates. Likewise, the use of broad or non-anatomically specific chlamydia codes (e.g., “other

chlamydial diseases”) may misclassify non-STI infections. Please review and refine the code lists where appropriate, and clearly acknowledge these limitations.

We appreciate the limitations associated with codes for recording these outcomes and have clarified several steps undertaken to mitigate risks identified. For syphilis, we have clarified that we required diagnoses to occur within 10 days of a penicillin injection and ≥ 365 days after any previous diagnosis to mitigate risk that codes captured historical or duplicate diagnoses (lines 292-293). For gonorrhea and chlamydia, we have clarified that we included only diagnoses occurring ≥ 30 days after a previous diagnosis (lines 291-292).

We have also added columns to Table S23 (gonorrhea codes), Table S24 (chlamydia codes), Table S25 (syphilis codes) indicating the frequency with which each code was recorded in the dataset. We have added a corresponding column to Table S26 (procedure codes for injections) recording the frequency with which these were assigned in association with an STI diagnosis. Reassuringly, the frequencies reported in these columns confirm that non-standard treatments and ambiguous diagnosis codes occur very infrequently. Specifically,

- **For chlamydia, <0.1% of all diagnoses were associated with the “Other chlamydial diseases” (A74.8/A74.81) codes. Overall, >98% of diagnoses are coded with either “Chlamydia infection, unspecified” (A74.9) or codes specific to genitourinary, anal/rectal, pharyngeal chlamydia infections or those designated as sexually-transmitted.**
- **For injections, macrolide injections accounted for only 0.6% of all injections occurring within the eligible time window around STI diagnoses, and <0.1% of all STI episodes captured in our study involved such treatment.**

Based on these observations, we believe any quantitative impact of these codes is negligible; excluding them would not change numerical results at the level of precision reported in the manuscript and tables/figures. We appreciate the opportunity to add the individual code frequencies and hope this helps to provide further context about both the STI outcomes and antibiotic treatment practices studied. We have expanded the limitations paragraph to address the points identified, with which we agree (lines 229-231).

- The analysis does not appear to account for on-demand (event-driven) PrEP use, which is increasingly common. By equating "active" or "consistent" PrEP use with ongoing sexual risk and likely doxyPEP candidacy, the model may overestimate doxyPEP frequency among those with intermittent risk. Please acknowledge this in the manuscript and, if feasible, discuss implications for the model's projections.

As our analysis period covers PrEP use from 2015-2019 we believe daily use is the primary modality represented in the patient population studied; the first guidelines for event-driven PrEP emerged late in this period (Saag et al., *JAMA* 2018 for the International Antiviral Society—USA panel, and 2019 for the European AIDS Clinical Society). However, event-driven use may have already been occurring and represents an important consideration, especially for the current context. Although we cannot distinguish use patterns from claims data alone, we believe both daily and event-driven use are represented. Codes used to capture emtricitabine/tenofovir use are not restricted by the number of doses contained in any fill. Thus, even in the event that individuals who intend to use PrEP on an event basis receive smaller drug volumes, they are not excluded from the primary analyses encompassing all PrEP users.

Of two use patterns that we subclassified for additional analyses, we believe that “Active” use imposes no discrimination on event-driven versus daily PrEP: this simply refers to any period within 3 months after a fill, which we have clarified in this revision (lines 283-288).

“Consistent” use is defined as any period preceded by 12 months during which individuals received ≥ 3 fills. We recognize some individuals using event-driven PrEP may be excluded from the “consistent” use category, particularly if they do not engage in HIV/STI testing at 3-6 month intervals or do not receive refills in conjunction with testing at this cadence. Since most fills contain 90 doses, “consistent” use would represent 270 doses or the amount of drug needed for 67 event-driven (2-1-1) uses—an average of one event every 5.5 days if individuals are exhausting their supply before receiving a refill. However, as PrEP recipients are recommended to receive HIV/STI testing every 3-6 months regardless of whether they have exhausted their pill supply, and likely receive refills in association with testing, we note that some individuals using PrEP under event-driven patterns could plausibly still receive ≥ 3 fills in a 12-month period. We have revised the Methods (lines 283-288) to clarify these considerations and address that

event-driven PrEP users would be misclassified as non-users only if they received 0 prescriptions in the preceding 12 months—a circumstance that may indeed suggest disengagement from PrEP use.

We believe this issue does not impact our estimates of doxyPEP frequency, however, as our framework for estimating doxycycline consumption is wholly decoupled from PrEP use patterns. We used data on sexual encounter frequency within the ARTnet study (Methods, lines 315-321 and Text S1) to estimate the frequency of doxyPEP use, which is not specific to daily versus event-driven PrEP use. Although we did estimate avertible incidence during periods where individuals met criteria for “active” or “consistent” PrEP use, it is important to note these did not differ appreciably from rates for periods with “any” PrEP use (≥ 1 prescription in 12 months). “Any” PrEP use was associated with 10.8 STI episodes/100 person years (vs. 11.5-13.0 for active/consistent use; Table 1), and estimated increases in tetracycline use were nearly identical across these categories (203.5/100 person-years for any PrEP use and 203.4/100 person-years for both “active” and “consistent” use periods; Table 3). Collectively, these factors suggest our results are not heavily influenced by daily versus event-driven PrEP use patterns.

- The assumption that doxyPEP uptake will mirror PrEP use—both in terms of population and adherence—is not explicitly justified. While PrEP users are likely early adopters, real-world data suggest some doxyPEP users are not on PrEP or have different behavioral profiles. I recommend acknowledging this assumption and supporting it with relevant citations or qualifying its limitations in the modeling framework.

We agree and have revised the Discussion to reflect this consideration, citing data on current or prior PrEP use among doxyPEP adopters in recent real-world studies (lines 196-201).

Results

• Figures 1 and 2 (pp. 15–16) clearly and effectively illustrate the modeled trade-offs between STI prevention and antibiotic use across subgroups.

We are pleased to know these trade-offs come through in the figures and thank the Reviewer for this comment.

Discussion

• Please clarify the implications of declining doxyPEP effectiveness for gonorrhea over time due to emerging tetracycline resistance. While Table S15 provides a sensitivity analysis assuming no effect on gonorrhea, this important scenario deserves more discussion, particularly given that gonorrhea reduction appears to be the largest driver of benefit.

We have added a paragraph to our Discussion to address the proportion of all preventable burden associated with gonorrhea, and to review evidence on the lack of benefit of doxyPEP in settings with prevalent tetracycline resistance in *N. gonorrhoeae*. We also address emerging evidence about resistance selection in *N. gonorrhoeae* associated with doxyPEP use, and implications for other antibiotics as resistance across drug classes is often linked (lines 212-218).

• The study period (2016–2019) overlaps with a time of increasing PrEP use, declining condom use, and rising STI rates. These secular trends suggest that modeled estimates may underestimate today’s STI incidence and avertible burden. The authors should acknowledge that their findings may be conservative relative to current dynamics and that newer behavioral data could further inform implementation projections.

We have added these points to the Discussion as suggested (lines 236-238).

• The manuscript notes disparities in doxyPEP awareness and use (e.g., among uninsured individuals, rural MSM), but stops short of exploring how the findings might inform equitable implementation. I encourage the authors to expand on how their results could guide more equitable policy design, such as integrating doxyPEP into public STI clinics or ensuring access among underserved populations.

We have revised the Discussion to address these specific considerations regarding equitable access (lines 246-248).

Reviewer #2 (Remarks to the Author):

The analysis is timely and adds important context for the discussion on benefits and disadvantages of doxyPEP

- Abstract: please include information on the type of model used

We have revised the Abstract to clarify model accounted for estimates of doxyPEP prophylactic efficacy against each STI studied and was applied to the estimated rates of diagnoses from our first-stage analysis of incidence among differing risk groups (lines 33-34).

- Observed diagnosis rate is not the same as unobserved incidence rate, and referring to the outcome of interest consistently as diagnosis rates would improve clarity

We agree with this suggestion and have revised the text to refer to incidence rates as “STI diagnoses” rather than total rates of STIs, which would encompass both observed and unobserved infections. The one exception is in lines 61-62 of the Introduction and lines 316-317 of the Methods, where we describe a trial that included active monitoring for infections (thus more likely capturing true incidence of infection).

- This above difference is significant also for interpretation of the results. Not all incident infections are diagnosed, and key estimates in this study quantify increases in antibiotic use compared to diagnosed and treated infections. The analysis does not capture the number of infections averted, in people using doxyPEP, that would not have been diagnosed. While this would not change the conclusions of the study when using diagnosis rate as the unit of measure, I would encourage you to discuss the full range of potential direct benefits of doxyPEP, and limitations of the data in capturing these.

We agree and have added this consideration to the Discussion (lines 221-222, 224-227) in the context of preventing infections that may otherwise be detected only through STI screening, which may occur after individuals have unknowingly transmitted infection to their partners.

- A brief explanation for conversion rates for different antibiotic modalities (treatment vs doxyPEP) is helpful; consider placing this in the main-text for clarity.

We have included this explanation in the main text at first mention (lines 121-124); we have also kept the additional description in the Methods.

- The abbreviation IRD is defined in a figure, but not in the method text (methods line 311).

We thank the Reviewer for catching this omission and have added the abbreviation where the term first appears in the Methods (line 332).